# Advances in the Technologies for Marine Salinity Measurement

Lijuan Gu [1,2,3], Xiangge He [1,2,3,*], Min Zhang [1,3] and Hailong Lu [1,2]

1    Beijing International Center for Gas Hydrate, Peking University, Beijing 100871, China
2    School of Earth and Space Sciences, Peking University, Beijing 100871, China
3    Dongguan Institute of Optics and Electronics, Peking University, Dongguan 523000, China
*    Correspondence: hexiangge@pku.edu.cn

**Abstract:** As one of the most important physical parameters of seawater, salinity is essential to study climatological change, to trace seawater masses and to model ocean dynamics. The traditional way to conduct salinity measurement in hydrographical observation is to use a standard conductivity-temperature-depth (CTD) probe where the salinity determination is based on a measurement of electrical conductivity. This article describes some developments of recent years that could lead to a new generation of instruments for the determination of salinity in seawater. Salinity determination with optical salinity sensor based on the refractive index measurement have been extensively studied. Different ways to conduct refractive index measurements are summarized, including measurements based on beam deviation, light wave mode coupling and swelling of surface coating material, among which the optical fiber sensors are promising candidates for further commercialization. Complementary to the above-mentioned direct measurement salinity point sensors, seismic observation takes advantages of large scale multichannel seismic data to retrieve the ocean salinity with high lateral resolution of ∼10 m. This work provide comprehensive information in the techniques related to the marine salinity measurement.

**Keywords:** salinity measurement; CTD; refractive index; optical fiber salinity sensor; multichannel seismics

## 1. Introduction

Salinity is one of the physical parameters of seawater [1]. Salinity is important to climatological observation [2], and it is also used to trace seawater masses and to model ocean dynamics [3–6].

The ocean salinity is defined as "the total amount of solid material in grams contained in one kilogram of seawater when all the carbonate has been converted to oxide, all the bromine and iodine replaced by chlorine, and all the organic material oxidized" [7]. Sea water salinity was determined in the early 20th century using the chlorinity and the empirical Knudsen relation. Since the chlorinity can be accurately measured, while it is not true for salinity [8]. Apart from chlorinity, conductivity has also served as a measure of salinity with formulas of the PSS-78 [9]. In the formulation of PSS-78, the salinity refers to the practical salinity defined in terms of the electrical conductivity ratio of the seawater sample to a reference solution at certain temperature and pressure. Thus, the practical salinity is a unit-less quantity and is commonly adopted in practical applications, since it can be calculated directly from the measured conductivity, temperature and pressure. Typically, seawater has a salinity around 35, and the salinity of coasts and rivers near the ocean are lower than that [10]. In 2009, the absolute salinity expressed in units of g/kg, is defined in the document entitled the Thermodynamic Equation of Seawater-2010 or TEOS-10, as the official description of seawater in marine science [11]. The absolute salinity is defined as the mass fraction of dissolved material in seawater. The absolute salinity provides the best estimate of the density of seawater. Because all the salinities stored in oceanographic databases are practical salinities, and to keep the compatibility between the

requirements of the TEOS-10 and the databases contains, a concept of reference salinity $S_R$ has been defined [8].

With the commercialization of electrical conductivity sensors in 1970s, in-situ measurement of the ocean salinity is available with the formulas of the PSS-78 that provide the relationship between electrical conductivity and salinity [9]. With the conductivity from the electrical conductivity sensors, practical salinity at certain temperature and pressure can be determined. Plenty of high performance conductivity sensors have been developed and some of the representative products are summarized in this paper. Challenges remain to produce small and low power conductivity sensors that are satisfying the high demands on measurement uncertainty needed for oceanographic research.

Another way to assess the seawater salinity is the optical sensors based on the refractive index measurement [12,13]. The seawater can be considered as an optical medium. The refractive index is directly related to density of seawater through the Lorentz-Lorenz relation. Meanwhile, the density changes with the salinity and temperature, and the refractive index also changes with wavelength. Thus salinity of seawater can be determined with its refractive index, here the salinity refers to the absolute salinity. There are different ways to conduct refractive index measurements, including measurements based on beam deviation, light wave mode coupling and surface coating. Great efforts have been made of the use of optical refractometers in situ, while obstacles stay to make of them instruments able to challenge conductivity cells in salinity resolution and precision. Up until now, the salinity measurement resolution of optical sensors is generally lower than that of conductivity sensors.

Oceanic hydrography deals with the measurement and description of the physical features of oceans and their evolution [14,15], as well as understanding of the circulation on the various temporal and spatial scale, for the purposes of navigation safety, scientific research, environmental protection, security and defense [16,17]. During the oceanic hydrographical survey, the electronic and optical salinity sensors are typically deployed from a platform at a certain location and conduct salinity measurement at different water depths. The vertical precision of the measurement can achieve as high as $\sim$1 m. While the measurement locations are always distant with each other, as a result, the salinity distribution of seawater in the lateral direction is poorly determined. On the other hand, salinity measurement with seismic observation has a higher spatial sampling in horizontal resolution of $\sim$10 m. Seismic observation takes advantages of the large-scale seismic observations to inverse the physical properties of the seawater including temperature and salinity with high lateral resolution. Thus, it is indirect measurement of marine salinity and can be applied to investigate various physical phenomena including internal waves, thermohaline intrusions and so on [18].

In this paper, the ocean salinity sensors with mature electronic instruments and the research development in this area are summarized in Section 2. Then the refractive index based optical instruments are summarized in Section 3. The seismic observation technique based on different inversion strategies to retrieve salinity of the seawater is described in Section 4. Finally, a conclusion is made.

## 2. Salinity Determination Using Electrical Conductivity Sensors

### 2.1. Measuring Principle of Electrical Conductivity Sensors

The electrical conductivity sensors can be divided into two groups: conductive conductivity sensors containing two or more electrodes and inductive conductivity sensors containing one or two transformers [19]. The electrical conductivity sensors obtain the salinity value by measuring the electrical conductivity of seawater. Since the electrical conductivity is also dependent on temperature and pressure, it is necessary to obtain the values of temperature and pressure at the same time. The salinity can be calculated as [9,20]:

$$S = 0.0080 - 0.1692R_T^{1/2} + 25.3851R_T + 14.0941R_T^{3/2} - 7.0261R_T^2 + 2.7081R_T^{5/2}$$
$$+ \frac{(T-15)}{1+0.0162(T-15)}(0.0005 - 0.0056R_T^{1/2} - 0.0066R_T - 0.0375R_T^{3/2} + 0.0636R_T^2 - 0.0144R_T^{5/2}) \tag{1}$$

where $S$ is the practical salinity, $T$ is the temperature in Celsius degree (°C) and $p$ is the pressure in decibar (dbar). $R_T$ is a function of both temperature and salinity and $-2\,°\mathrm{C} \leq T \leq 35\,°\mathrm{C}$. $R_T$ can be calculated as:

$$R_T = \frac{R}{R_p \cdot r_T} \tag{2}$$

where $R = C(S,T,p)/C(35,15,0)$ with $C(S,T,p)$ representing the electrical conductivity of ambient water of salinity $S$ at temperature $T$ and pressure $p$ and $C(35,15,0) = 4.2914\,\mathrm{S/m}$ representing the electrical conductivity of standard seawater of practical salinity 35, at 15 °C and atmospheric pressure. It should be noted that the stated conductivity of 4.2914 S/m is not part of the definition of the practical salinity scale PSS-78. It is stated here for informational purposes only. $R_p = C(S,T,p)/C(S,T,0)$ is the pressure correction of conductivity and its empirical equation can be expressed as:

$$R_p = 1 + \frac{2.070 \times 10^{-5}p - 6.370 \times 10^{-10}p^2 + 3.989 \times 10^{-15}p^3}{1 + 3.426 \times 10^{-2}T + 4.464 \times 10^{-4}T^2 + 4.215 \times 10^{-1}R - 3.107 \times 10^{-3}TR} \tag{3}$$

$r_T = C(35,T,0)/C(35,15,0)$ is the temperature correction of conductivity, and its empirical equation can be expressed as:

$$r_T = 6.766097 \times 10^{-1} + 2.00564 \times 10^{-2}T + 1.104259 \times 10^{-4}T^2 - 6.9689 \times 10^{-7}T^3 + 1.0031 \times 10^{-9}T^4 \tag{4}$$

Then we use Equations (1)–(4) to show the relation of salinity and conductivity, temperature and pressure. The results are shown in Figure 1. Figure 1a–c shows the volume slice planes of pressure (Figure 1a), conductivity (Figure 1b) and temperature (Figure 1c), respectively. In one of the volume slice plane of pressure, for example, a colored contour map of salinity in the plane of temperature and conductivity on that slice is intuitively shown. At a constant pressure, variation in temperature induce a density variation and the dissociation constant. This changes the number of ions per unit volume as well as the viscosity. The measured conductivity varies, thus the salinity changes [2]. Figure 1d–f shows the relationship between salinity with conductivity, temperature and pressure, keeping the other two variables at constant values. For instance, Figure 1d shows the salinity versus conductivity under different temperature and pressure. It can be seen from the figure that temperature has the greatest impact on salinity and the pressure hardly affect the salinity.

As can be seen from Figure 1d, the relationship between salinity and conductivity is almost linear. Table 1 shows the sensitivity between salinity and conductivity under different temperature and pressure. The data fit excellently with a linear function with all $R^2$ larger than 0.999. With decreasing temperature at constant pressure, the salinity increase more sharply with conductivity. Thus the temperature has a great influence on the salinity measurement results. While under different pressures, the sensitivity of salinity with conductivity hardly changes, indicating that the pressure has little effect on the salinity measurement.

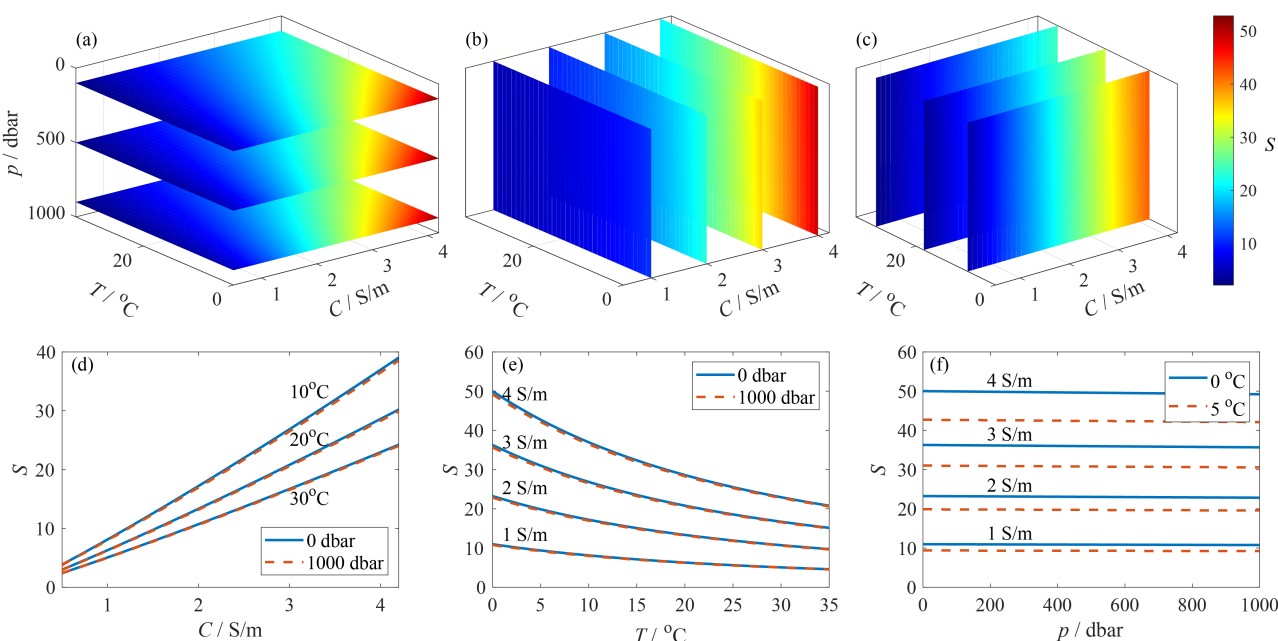

**Figure 1.** The relation of salinity and conductivity, temperature and pressure. (**a**) Pressure slice planes. (**b**) Conductivity slice planes. (**c**) Temperature slice planes. (**d**) The salinity versus conductivity under different temperature and pressure. (**e**) The salinity versus temperature under different conductivity and pressure. (**f**) The salinity versus pressure under different conductivity and temperature.

**Table 1.** The ratio of change of salinity $\Delta S$ to change of conductivity $\Delta C$ at different temperature-pressure points.

| Temperature/°C | Pressure/dbar | $\Delta S/\Delta C$/m/S | $R^2$ |
|:---:|:---:|:---:|:---:|
| 10 | 0 | 9.535 | 0.9993 |
| | 1000 | 9.418 | 0.9992 |
| 20 | 0 | 7.376 | 0.9993 |
| | 1000 | 7.304 | 0.9993 |
| 30 | 0 | 5.915 | 0.9993 |
| | 1000 | 5.868 | 0.9993 |

### 2.2. The CTD Instruments

The Conductivity-Temperature-Depth (CTD) sensor is one of the most used instruments in the Oceanographic field [21,22]. It is generally called the thermohalimeter. The CTD instrument is used to measure the three basic physical parameters of water body, namely conductivity, temperature and pressure [22]. With these parameters, other physical parameters of the seawater, such as depth, salinity and sound speed, can be calculated.

CTD instruments have been developed to maturity, Table 2 lists the CTD instruments of different companies and their conductivity measurement capabilities. As summarized in Table 2, high-accuracy CTD sensors are present on the market and widely used. The current research in this area is mainly towards the miniaturization and integration of CTD, such as the animal-mounted instruments that can simultaneously recording movements, diving behavior, and in situ oceanographic properties [23].

**Table 2.** The performance of commercial CTD instruments.

| Country | Company | Product Name | Measuring Range (S/m) | Resolution (S/m) | Accuracy (S/m) | Response Time | Stability |
|---|---|---|---|---|---|---|---|
| Canada | AML Oceanographic | X2change Sensor [24] | 0–9 | 0.0001 | ±0.001 | 25 ms | n/a [1] |
| Canada | RBR Ltd. | RBRmaestro Multi-Channel Logger [25] | 0–8.5 | 0.0001 | ±0.0003 | n/a | 0.001 S/m per year |
| France | nke Instrumentation | WiMo: multiparameter sonde [26] | 0–10 | 0.00001 | ±0.05 | n/a | n/a |
| Germany | Sea & Sun Technology GmbH | CTD 115 M [27] | 0–7 | 0.00005 | ±0.0002 | 150 ms | n/a |
| Italy | Idronaut S.R.L. | Ocean Seven 320 Plus WOCE-CTD [28] | 0–7 | 0.00001 | ±0.0001 | 50 ms | n/a |
| Japan | JFE Advantech Co., Ltd. | AAQ-RINKO, profiler [29] | 0.05–7 | 0.0001 | ±0.001 | 0.2s | n/a |
| Norway | Aanderaa Data Instruments | SEAGUARD CTD [30] | 0–7.5 | 0.0002 | ±0.0018 | <3 s | n/a |
| United Kingdom | Valeport | MIDAS CTD+ [31] | 0–8 | 0.0002 | ±0.001 | n/a | n/a |
| USA | Sea-Bird Scientific | SBE 911plus CTD [32] | 0–70 | 0.0004 | ±0.003 | n/a | 0.003 S/m per month |
| USA | SonTek | CASTAWAY-CTD [33] | 0–10 | 0.0001 | ±0.025 | n/a | n/a |

[1] Not available.

Micro-Electro-Mechanical System (MEMS) is a mature technology that is beneficial to the miniaturization and batch fabrication of the electronic salinity sensor [6,34–36]. Hyldgård et al. designed and fabricated a micro CTD system for ocean water salinity measurement [6]. The chip layout of the sensor is shown in Figure 2a that contains a piezoresistive pressure sensor (p), a thermistor temperature sensor (t) and four conductivity electrodes (c) for determination of the salinity of the water. As illustrated in Figure 2b, the chip is as compact as 4 mm × 4 mm. According to the measured trans-impedance as a function of frequency in Figure 2c, there exists a frequency interval (named salinity window) in which the impedance, hence the conductivity, is highly dependent of salinity. Finally, the frequency of 3.4 kHz was used for the conductivity sensor. The conductivities of the water samples were measured with an accuracy of 0.06 S/m and the temperature and pressure were measured with an accuracy of ±0.13 °C and ±0.5 dbar, respectively. The salinity can therefore be detected with an accuracy of ±0.5. Although the accuracy of the sensor is not high, it is sufficiently for being mounted on the fish to reconstruct fish migration and patterns of fish behaviour. Wu et al. investigated the feasibility of batch microfabricated Conductivity-Temperature (CT) sensors using MEMS technology for marine measurements [34]. The fabricated 34 CT sensors showed excellent consistency of ±0.00048 S/m, as well as high conductivity precision of ±0.002 S/m, accuracy of ±0.008 S/m, and sensitivity of 2.5 cm$^{-1}$. The results indicate that the batch microfabricated sensors are suitable for a large-scale deployment in the Marine Internet of Things.

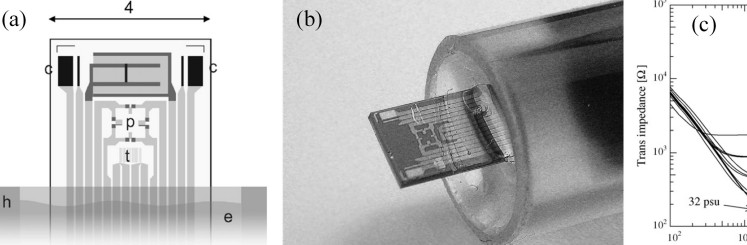

**Figure 2.** The micro CTD system for measurement of ocean water salinity. (**a**) The chip layout. (**b**) Photograph of finished chip. (**c**) Measured trans-impedance as a function of frequency for five different salinities (2, 4, 8, 16 and 32 psu). Reprinted from Publication [6], Copyright (2022), with permission from Elsevier.

Aravamudhan et al. proposed a MEMS based silicon CTD sensor for ocean environment and the sensor elements were carefully protected from the external fields [35]. The CTD sensor can provide an accuracy of <0.5% for depth, 0.02% for conductivity and 0.05 °C for temperature. Ashokan et al. proposed four electrode conductivity cell configuration to shield the effect from external electric field that affect the inductive sensors for ocean water salinity measurement [36].

In cases demanding large numbers of CTD sensors, an inexpensive version is needed, especially if used as a one-time system. Broadbent et al. fabricated the low cost and expendable CTD sensor with novel printed circuit board (PCB) MEMS techniques [37,38]. The conductivity cell had an estimated precision of 0.0032 S/m, an estimated accuracy of ±1.47% and a detection range of 0 to 6 S/m. Here the accuracy was estimated to calculate the mean absolute percentage error (MAPE) by summing the absolute difference between the two sensors, dividing by the sum of the reference readings and multiplying by 100%. The estimated temperature precision, accuracy and detection range were 0.0066 °C, ±0.546 °C and 0 to 50 °C. The same group further designed a low-cost, compact CTD biologger for medium-sized marine animals adopting the PCB MEMS techniques [39]. The miniaturized conductivity sensor can be integrated with the commercial thermistors and pressure sensors. The sensitivity and precision of the conductivity sensor were 0.8–1.0 cm$^{-1}$ and ±0.014 S/m in the range of 0.2–7 S/m. The piezoresistive pressure module showed an accuracy of ±0.2 dbar and a range of 0 to 140 m. Paradis et al. described the design, development, construction and testing of the Expendable Conductivity Sensor Unit [21]. The sensor unit is tested to meets all the requirements for this instrument, including light weight, compact size, reasonably low cost, and wireless communication.

The development of graphene-based sensors is prompted because of its outstanding mechanical, electrical, and thermal properties. A new type of salinity sensor based on laser-induced graphene was designed and fabricated [40,41]. These sensors have the advantages of low cost, corrosion resistance and bio-compatibility [40].

## 3. Salinity Determination Using Optical Sensors

### 3.1. Measuring Principle of Optical Sensors

The seawater is considered as an optical medium and the optical instruments obtain the salinity value mainly by measuring the refractive index of the seawater. The refractive index is directly related to density $\rho$ of seawater through the Lorentz-Lorenz relation. Meanwhile, the density changes with the salinity ($S$) and temperature ($T$), and the refractive index also changes with wavelength ($\lambda$). As a result, the salinity can be determined with refractive index. Several authors devoted to determining the empirical relationship of $n(S, T, \lambda)$ [42,43]. Millard and Seaver developed a 27-term index of refraction algorithm for pure water and sea waters in accordance to the four experimental data sets [42]. The accuracy of the algorithm varying with pressure, with 0.0004 at atmosphere pressure and 0.08 at high pressures. The equation covers the range 500–700 nm in wavelength, 0–30 °C in temperature, 0–40 in salinity and 0–11,000 dbar in pressure. Quan and Fry determined a

simple ten-parameter empirical equation for the refractive index as a function of salinity, temperature and wavelength at ambient pressure that can be expressed as [43]:

$$n(S, T, \lambda) = 1.31405 + (1.779 \times 10^{-4} - 1.05 \times 10^{-6}T + 1.6 \times 10^{-8}T^2)S - 2.02 \times 10^{-6}T^2$$
$$+ \frac{15.868 + 0.01155S - 0.00423T}{\lambda} - \frac{4382}{\lambda^2} + \frac{1.1455 \times 10^6}{\lambda^3} \tag{5}$$

where $S$ is the salinity, $T$ is the temperature in Celsius degree and $\lambda$ is the wavelength in nanometer (nm). The ranges of validity are 0 °C $< T <$ 30 °C, $0 < S < 35$, and 400 nm $< \lambda <$ 700 nm. This equation can reproduce the experimental data within the experimental errors.

We use Equation (5) to show the changes of refractive index with changing salinity, temperature and wavelength. The volume slice planes of temperature (Figure 3a), salinity (Figure 3b) and wavelength (Figure3c) are shown in Figure 3. Figure 3d–f shows the relationship between refractive index with salinity, temperature and wavelength, keeping the other two variables at different constant values. For instance, Figure 3d shows the refractive index versus salinity under different temperature and wavelength.

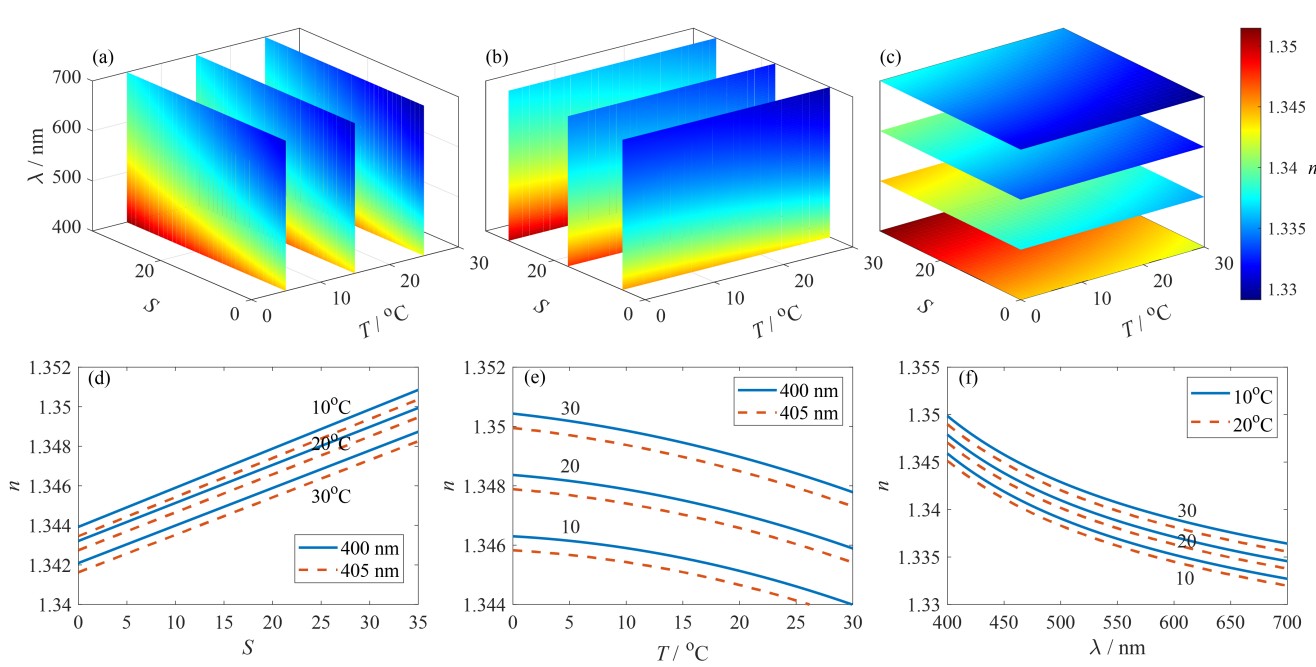

**Figure 3.** The relation of refractive index and salinity, temperature and wavelength. (**a**) Temperature slice planes. (**b**) Salinity slice planes. (**c**) Wavelength slice planes. (**d**) The refractive index versus salinity under different temperature and wavelength. (**e**) The refractive index versus temperature under different salinity and wavelength. (**f**) The refractive index versus wavelength under different salinity and temperature.

As can be seen from Figure 3d, refractive index increases linearly with salinity at different temperatures and wavelengths. This relationship is quantified in Table 3 as the ratio of change of salinity $\Delta S$ to change of refractive index $\Delta n$ at different temperature-wavelength points. The data fit excellently with a linear function with all $R^2$ equals to 1. The ratio $\Delta S / \Delta n$ varies at different temperature-wavelength points, indicating that the temperature and wavelength have a significant influence on the measurement results.

**Table 3.** The ratio of change of salinity $\Delta S$ to change of refractive index $\Delta n$ at different temperature-wavelength points.

| Temperature/°C | Wavelength/nm | $\Delta S/\Delta n$ | $R^2$ |
|:---:|:---:|:---:|:---:|
| 10 | 400 | 5054 | 1 |
| | 405 | 5063 | 1 |
| 20 | 400 | 5204 | 1 |
| | 405 | 5213 | 1 |
| 30 | 400 | 5272 | 1 |
| | 405 | 5282 | 1 |

From the above analysis, it can be seen that the refractive index has a linear relationship with the salinity of the solution. The refractive index can be directly measured by optical means. But to obtain the salinity, we also need to know the temperature and wavelength. Compared with the mature application of electronic instruments, there are only a few optical measurement products in the market, such as the NOSS sensor from NKE [25], and other optical measurement methods are still in the research stage. Various optical measurement methods have been proposed, including measurement based on beam deviation, light wave mode coupling and swelling of surface coating material. The sensor configuration, working principle and measurement performance of these methods are reviewed in detail in the following. Since optical sensors are mainly in the research stage, this review is mainly for scientific researchers, and the measurement principle of different sensors are reviewed respectively.

*3.2. Salinity Measurement Based on Beam Deviation*

Figure 4 shows the principle of salinity measurement based on beam deviation. A partitioned cell is used to contain the reference solution and the sample solution. The incident angle of light at the interface of these two solutions is $\alpha$ and the refraction angle is $\beta$. The change of the salinity, hence the refractive index, of the sample solution will lead to the deviation of beam propagation direction. By measuring the beam deviation with a position-sensitivity detector (PSD), the salinity of the solution can be obtained.

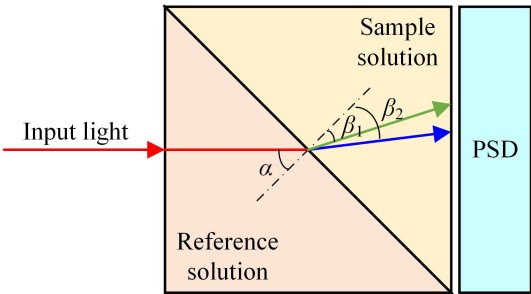

**Figure 4.** Principle of salinity measurement based on beam deviation.

The beam deviation method can be divided into transmission type and reflection type. As early as 1989, Minato et al. presented a salinity sensor based on transmission type refractometer[13]. In their system, a partitioned cell was designed which was divided into two parts: one contained standard seawater (the salinity is 35) for reference, and the other was filled with sample seawater. When the salinity, hence the refractive index, of the sample seawater was different from that of the reference seawater, the beam propagating in the cell will deviate, and the refractive angle change was nearly proportional to the salinity. In this system, the salinity measurement resolution was 0.02 with an accuracy of $\pm 0.6$ from salinity range of 0 to 40. In 2003, Zhao et al. described a salinity sensor based

on the measurement of the beam deviation due to the refractive angle change [44]. In this system, the beam from a light source transmitted through a single-mode fiber before entering the sensor probe. The deviated light beam caused by water with different salinity entered the receiving fibers (40 multi-mode fibers arranged in linear array), and was then measured by the position-sensitive detector (PSD). The sensor probe mainly composed of two parts: the sample water tank containing a wedge-shaped reference cell filled with distilled water and the right-angle prism used to reflect the incident beam. The incident beam will then sequentially pass through the reference water, the oblique plate, the sample water, and then be reflected by the right-angle prism. When the salinity of the sample water changes, the deviation of the beam changes synchronously. Based on the transverse photoelectric effect, the PSD's output signal was independent to the incident light intensity and only related to positions of the incident beam. The salinity measurement resolution was 0.012 with a salinity accuracy of 0.28 within the measurement range from distilled water to salt water with salinity of 50. It is worth to mention that the NOSS sensor from NKE is a refractometer with reflection type. The beam path has been deviated by gold mirrors deposited on angles of the prisms specially sized to reflect the beam on the PSD, making integration in a container easier [45]. The salinity measurement accuracy is $\pm0.005$ within the measurement range 15 to 42 [25].

### *3.3. Salinity Measurement Based on Light Wave Mode Coupling*

The salinity sensor based on light wave mode coupling typically obtain the salinity by means of the light transmission wave variation. The optical transmission medium is surrounded by the solution of which the salinity is to be quantified. Since the light wave interact with the surround medium, the intensity or the spectrum of the transmission wave changes with the refractive index of the surround solution. Thus high accuracy of salinity measurement can be achieved with light wave mode coupling for refractive index measurement. The light wave mode coupling method can be divided into light intensity detection and wavelength detection.

Figure 5a shows the schematic diagram of salinity measurement based on light intensity detection. The sensor probe is immersed in the sample solution, and the light emitted by the light source reaches the detector through the optical fibers and the sensor probe. The evanescent field of the guided mode will extend into the solution. Different salinity leads to different coupling efficiencies of the incident light field into the solution, so the detection light intensity varies. The sensor probe can be realized in many ways, such as surface-plasmon resonance, tapered optical fiber or U-shaped plastic optical fiber. In these sensor probes, some light will radiate into the solution as the evanescence wave, and the variation of the salinity in water will lead to the change of the light intensity in the fiber. In 1999, Esteban et al. used a fiber-optic sensor based on surface-plasmon resonance to measure the salinity of water [46]. The sensor probe consisted of a metallic layer deposited on a side-polished single mode optical fiber. A part of the evanescent field of the guided mode is coupled as a surface plasmon in the metallic layer. This effect reduced the power transmitted by the fiber and the attenuation was strongly dependent on the refractive index, hence the salinity, of the external solution that was in contact with the metallic layer. The measurement accuracy was $\pm0.1$ in the salinity range from 0 to 40. In 2011, Rahman et al. demonstrated a simple tapered plastic multi-mode fiber-optic sensor for monitoring of salinity [47]. The sensor probe was a tapered fiber which was fabricated using a heat-pulling method to achieve a waist diameter and length of 0.187 mm and 5 mm respectively. The tapered fiber was immersed in the solution during measurement. The increment of salinity concentration increased the refractive index of the solution, which in turn reduced the index difference between core and cladding of the tapered fiber and thus allows less lights to be transmitted. The salinity measurement resolution was 1.152 with a sensitivity of 0.00024 mV and a salinity accuracy of $\pm2.532$ when the solution concentration varies from 0 to 120. In 2012, Wang et al. directly used a plastic multi-mode optical fiber to measure salinity [48]. U-shaped and spiral-shaped plastic optical fibers were used as sensor

probes. Some light will radiate into the ambient solution of the sensor as the evanescence wave. Therefore, the variation of the salinity in water will lead to the change of the light intensity in the fiber. In their experiment, the sensitivities of the U-shaped sensor probe was 0.042 mV with a salinity accuracy of about $\pm7.14$ when the NaCl concentration varied from 0 to 350.

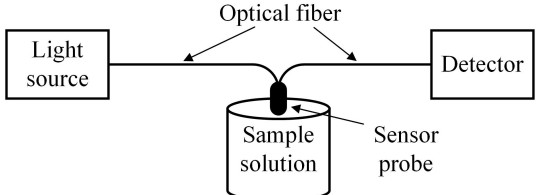

**Figure 5.** The schematic diagram of salinity measurement based on light intensity detection.

Although the salinity sensor based on intensity detection has a simple structure, the light intensity is easily affected by other factors, such as turbidity and bubbles, to limit its measurement accuracy. In order to improve the measurement accuracy, wavelength detection is used. Figure 6 shows the schematic diagram of salinity measurement based on wavelength detection. The sensor usually consists of lead-in optical fiber, sensing optical fiber and lead-out optical fiber. The lead-in and lead-out fibers are usually single-mode fibers (SMFs). The sensing fiber can be realized in many ways, such as two-core fiber, no-core fiber, or photonic crystal fiber, etc. The sensing fiber is exposed to the sample solution resulting in obvious dips on the output light spectrum, and the salinity can be obtained by tracking the shift of the dips. Therefore, the system setup of these sensors mainly consist of a broadband light source and an optical spectrum analyzer (OSA).

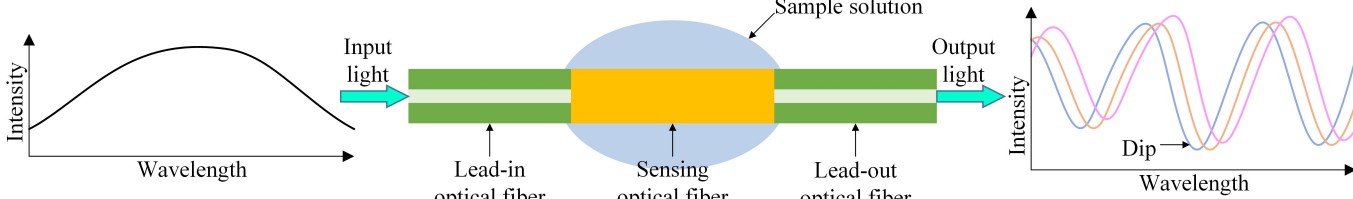

**Figure 6.** The schematic diagram of salinity measurement based on wavelength detection.

In 2013, Guzman-Sepulveda et al. demonstrated a highly sensitive salinity sensor based on a two-core optical fiber [49]. The sensor was fabricated by splicing a section of two-core fiber (TCF) between two single-mode fibers (SMFs). The central core was treated as the transmitting core while the off-axis core, the cladding around which was controllably removed by means of wet chemical etching, was treated as the coupling core. The TCF cladding around the off-axis core was etched using a solution of Hydrofluoric Acid known as Buffered Oxide Etchant (BOE), which slowly removes the cladding material thus allowing detecting the starting point of the interaction between the off-axis core and the surrounding liquid, i.e., when the spectral response starts to shift. The interaction with the surrounding media will induce changes on the effective refractive index of the off-axis core and this will manifest not only in variations on the coupling coefficient but also in a difference between the propagation constants, resulting in a spectral shift. There were obvious dips on the spectrum and the wavelength shift under different salinity can be obtained by tracking the dips. Measurement sensitivity of 14.0086 nm/(mol/L) was achieved for the salinity ranges from 0 to 5 mol/L. In 2014, Meng et al. proposed an optical fiber salinity sensor based on multi-mode interference (MMI) effect [50]. The sensor probe was manufactured by splicing a section of no-core fiber (NCF) between two single-mode fibers (SMFs), that possesses a transmission peak at a certain wavelength due to MMI effect. The input signal from the lead-in SMF entered into the NCF, where high order modes were

excited and MMI-induced self-imaging effect occurs. When the sensor was surrounded by salt solution, the external solution can be regarded as the cladding of the NCF. When refractive index of the salt solution increased, the effective refractive index of the NCF will be increased because the evanescent filed penetrates deeper into the liquid, and the effective diameter of the fundamental mode will be increased because the refractive index difference between the core and cladding was reduced. Thus, salinity of the liquid that it is immersed in can be measured from wavelength shift of the transmission peak. Linear response with salinity sensitivity of 1.94 pm was achieved and the measurement range was from 38.6 to 216.2. In 2019, Yu et al. demonstrated an all fiber refractive index sensor based on an optical microfiber coupler which was fabricated by fusing and tapering two twisted conventional communication fibers [51]. The output intensity of the coupler varied with light wavelength, thus forming a coupling spectrum that contained several dips. When external environment changed, the coupling spectrum will shift, and the measurement of salinity, temperature and depth (pressure) in seawater can be realized through monitoring the shift of the dips. The highest sensitivities of salinity, temperature, and depth were 1596 pm, 2326 pm/°C, and 169 pm/MPa respectively. In the same year, Wang et al. proposed an high-sensitivity salinity sensor with a exposed-core micro-structured optical fiber (ECF) based on a free space propagation-core mode Mach-Zehnder interferometer [52]. The proposed sensor had a salinity measurement range from 0 to 40 and a salinity sensitivity of around $-2.29$ nm.

In 2020, Mollah et al. theoretically proposed an ultrahigh sensitive photonic crystal fiber (PCF) salinity sensor based on the Sagnac interferometer (SI) [53]. In this sensor, all air holes were proposed to be filled with sea water. The refractive index of sea water variation induced birefringence changes reflected in the interference spectrum recorded by the spectrum analyser. Finite element method was used to analyze the propagation characteristics of the PCF. The achieved sensitivity was estimated to be 0.75 nm in the salinity range from 0 to 1000 (note that this is simulation data without practical significance) and the maximum salinity resolution was 0.133. Specially designed photonic crystal (PC) can also measure salinity. The PC is a multi-layer structure with a different dielectric constant in a periodic structure. As the incident light propagates through the PC, it is reflected at each interface. Under a suitable condition, constructive interference between the reflected waves occurs, and the resultant reflected wave destructively interferes with the incident wave. As a result, a forbidden band-gap appears as the resonant dip in the interference curve. In 2021, Zaky et al. theoretically studied a Tamm plasmon resonance-based one-dimensional PC sensor to calculate the salinity of seawater [54]. The sensor consisted of prism layer, Au layer, water layer, periodically arranged multiple Si layers and $SiO_2$ layers, and Si substrate in sequence. The resonant dip of the reflectance spectra varied with water salinity. The proposed sensor recorded a sensitivity of 1.432 nm. Besides, Akter et al. proposed a dual-core micro-structure optical sensor based on photonic crystal fiber [55] and Sayed et al. illustrated a two-dimensional photonic crystal salinity sensor [56]. However, these proposals are only verified by theoretical simulation, experimental confirmation should be conducted in the future.

### 3.4. Salinity Measurement Based on Swelling of Surface Coating Material

Another type of salinity measurement is based on mechanical stress that is induced in the chemically sensitive water swellable polymers coating when the water escapes from it caused by salinity increase. The mechanical stress acts upon the light wave that can be quantified through optical methods. In this method, optical measurement technology such as fiber Bragg grating (FBG), Sagnac interferometer (SI) and Fabry-Perot interferometer (FPI) are usually used, while the coating material usually includes hydrogel, epoxy and polyimide (PI), etc. In 2002, Cong et al. reported an optical salinity sensor using a fiber Bragg grating (FBG) coated with hydrogel [57]. Increase of the salinity of the surrounding solution will force water escaped from the hydrogel coating that induced mechanical stress in the coating. As a result, the stress in the hydrogel coating shifted the Bragg wavelength of the FBG. Salinity can be obtained by measuring the shift of the Bragg wavelength. In 2018,

Yin et al. reported an optical salinity sensor based on an optical microfiber coil resonator (MCR) with an epoxy coil [58]. When the salinity increased, the resonance wavelength will be red-shifted and the sensitivity can reach up to 1.5587 nm with a resolution of 0.0128. In addition to hydrogel and epoxy, another frequently used coating material is polyimide (PI). PI is an organic polymer material with good hygroscopicity, strength, anti-destructive and is environmentally friendly and safe. With these unique advantages, PI has been widely applied in optical fiber sensing areas in recent years. Once PI is dipped into distilled water, the polymer absorbs water and holds it. If salt is added to the distilled water, the net water concentration in the medium decreases that makes the PI layer to dissipate water out from it to shrinks the medium. Volume expansion and contraction of the PI layer are directly proportional to the concentration of water in the surrounding medium. In 2011, Wu et al. demonstrated a salinity sensor using a PI-coated photonic crystal fiber (PCF) Sagnac interferometer (SI) based on the coating swelling induced radial pressure on the PCF [59]. The achieved salinity sensitivity was 0.742 nm/(mol/L) with an accuracy of 0.027 mol/L. In 2015, Zhang et al. presented a fiber Fabry-Perot (FP) interference salinity sensor based on PI-diaphragm. During salinity increased from 0 to 5.47 mol/L, the maximum sensitivity was 0.45 nm/(mol/L) [60]. In 2019, Sun et al. developed a FBG salinity sensor coated with lamellar PI [61]. Compared with annular PI coating, lamellar PI coating enlarged the contact area with the solution so that more water molecules can be stored or released in the coating. The experimental results indicated that the sensitivity coefficients to water salinity was −0.00358 nm while the salinity accuracy was 1.048. In 2019, Kumari et al. demonstrated a salinity sensor based on an apodized FBG characterized by Nuttall profile coated with PI [62]. The results showed the sensor had a sensitivity of 0.0026 nm with an accuracy of 0.2015 in the range of 0 to 40.

In 2020, Zhang et al. demonstrated a distributed salinity sensor with a PI-coated polarization-maintaining photonic crystal fiber (PM-PCF) based on Brillouin dynamic grating (BDG) [63]. The PI-coating will swell or shrink if exposed to solutions of different salinity. The deformation to the coating were converted to the birefringence modulation loaded on the PM-PCF substrate. The salinity was then distributively measured by mapping the birefringence changes on the PM-PCF by BDG. The spatial resolution was 15 cm and the salinity sensitivity was 139.6 MHz/(mol/L) with a salinity accuracy of 0.072 mol/L.

The performance of the optical salinity sensors is summarized in Table 4. The given figures are converted to salinity to be comparable with the other sensors, such as 1 mol/L NaCl would be 58.44 g/L, so roughly a salinity of 58.44. Besides, the salinity accuracy is calculated as the ratio of the accuracy of the interrogator to the sensitivity of the proposed salinity sensor. The light wave mode coupling method is based on the change of guided wave mode with the solution concentration variation to measure salinity. Among them, sensors based on intensity detection have simple structures, while they are easily affected by external perturbations that limits their measurement accuracy. On the other hand, the sensors based on wavelength detection are less susceptible to interference, and high salinity accuracy can be obtained by extracting the wavelength interference dips. However, the fragile sensing head of different micro-structures fabricated with optical fiber shall be directly inserted into the solution, so proper packaging of the sensor is necessary in practical applications. Surface coated salinity sensors usually coat hydrogel, epoxy or polyimide material on the surface of optical fiber, and measure salinity through the transmission of mechanical stress. Since these coating materials have strong hygroscopicity, this kind of sensor can realize high sensitivity and good accuracy. However, for long-term applications, the stability of coating materials and the bonding strength between coating and optical fiber should be verified and improved.

**Table 4.** The comparison of various optical salinity sensors.

| Measuring Method | Configuration | Salinity Measurement Range | Sensitivity per Salinity Unit | Salinity Resolution | Salinity Accuracy | Ref. |
|---|---|---|---|---|---|---|
| Beam deviation | Transmission type refractometer cell (1989) | 0–40 | n/a [1] | 0.02 | ±0.6 | [13] |
| | Reflection type refractometer cell with a prism (2003) | 0–50 | n/a | 0.012 | ±0.28 | [44] |
| | NOSS sensor from NKE | 15–42 | n/a | n/a | ±0.005 | [25] |
| Light wave mode coupling | Surface-plasmon resonance optical fiber (1999) | 0–40 | n/a | n/a | ±0.1 | [46] |
| | Tapered plastic multi-mode optical fiber (2011) | 0–120 | 0.00024 mV | 1.152 | ±2.532 | [47] |
| | U-shaped plastic multi-mode optical fiber (2012) | 0–350 | 0.042 mV | n/a | ±7.14 | [48] |
| | Two-core optical fiber (2013) | 0–292 | 240 pm | n/a | ±0.0083 [2] | [49] |
| | No-core fiber multi-mode interference (2014) | 38.6–216.2 | 1.94 pm | n/a | ±1.03 [2] | [50] |
| | Optical microfiber coupler (2019) | 14.66–25.64 | 1596 pm | n/a | ±0.0013 [2] | [51] |
| | Exposed-core optical fiber (2019) | 0–40 | –2290 pm | n/a | ±0.00087 [2] | [52] |
| | Photonic crystal fiber Sagnac interferometer (2020) | 0–1000 [3] | 750 pm | n/a | ±0.0027 [2] | [53] |
| | Tamm plasmon resonance-based one-dimensional photonic crystal (2021) | 0–500 [3] | 1432 pm | n/a | ±0.0014 [2] | [54] |
| | Dual-core photonic crystal fiber (2020) | 0–700 [3] | 200 pm | n/a | ±0.01 [2] | [55] |
| Swelling of surface coating material | Optical microfiber coil resonator with epoxy coil (2018) | 34.6–35.6 | 1558.7 pm | n/a | ±0.0013 [2] | [58] |
| | Polyimide-coated photonic crystal fiber Sagnac interferometer(2011) | 0–292 | 12.7 pm | 1.58 | ±1.58 | [59] |
| | Fabry-Perot interferometer with Polyimide diaphragm (2015) | 0–320 | 7.7 pm | n/a | ±0.26 [2] | [60] |
| | FBG sensor coated with lamellar polyimide (2019) | 0–50 | −3.58 pm | 0.28 | ±1.048 | [61] |
| | Apodized FBG coated with poluimide (2019) | 0–40 | 2.6 pm | 0.115 | ±0.2015 | [62] |
| | Distributed polyimide-coadted photonic crystal fiber (2020) | 0–292 | 2.39 MHz | n/a | ±4.2 | [63] |

[1] Not available. [2] Estimated value when wavelength accuracy of the optical spectrum analyzer (OSA) is ±2 pm [49]. [3] Simulation data, without practical significance.

## 4. Salinity Measurement with Seismic Observation

Direct measurements of the physical properties of seawater by expendable bathythermographs (XBT) or conductivity-temperature-depth (CTD) profilers provide high vertical resolution of ∼1 m and poor horizontal resolution of ∼1 km. Multichannel seismic (MCS) is a widely used tool for imaging of the Earth's subsurface. The acoustic impedance of the seawater, represented with temperature, salinity and pressure by the equation of state, can be obtained with MCS. As a result, seismic observation of seawater salinity is a kind of

inversion problem. This new technique can sample the seawater with improved lateral resolution of ~10 m [64,65] and vertical resolution of ~10 m [66]. The ability of marine multichannel seismic reflection technique to retrieve the fine structures in the ocean show its promising prospects in oceanographic study [67].This promote the establishment of seismic oceanography (SO) that is the cross discipline between seismology and physical oceanography. SO can image various oceanic feathers including eddies, internal waves/solitary waves, etc. and quantitative inversion of the seawater parameters such as thermohaline structure [68] and salinity [18]. This paper focus on the salinity as well as temperature inversion from SO data and it should be noted that the contribution of salinity to the acoustic impedance of the seawater is about one-fourth of temperature [69].

### 4.1. Principle of Salinity Measurement from Seismic Data

The principle of multichannel seismic data collection is illustrated in Figure 7, the MCS data are acquired from a ship by firing an air gun array towed behind the ship. The towed streamer with a large number of equidistant distributed hydrophones, often several kilometers long and approximately 10 m beneath the sea level, record the reflected acoustic wave field. The effective horizontal resolution of this technique that determined by the wavelet peak frequency is ~10 m [64,65] and the vertical resolution is approximately 10 m [66].

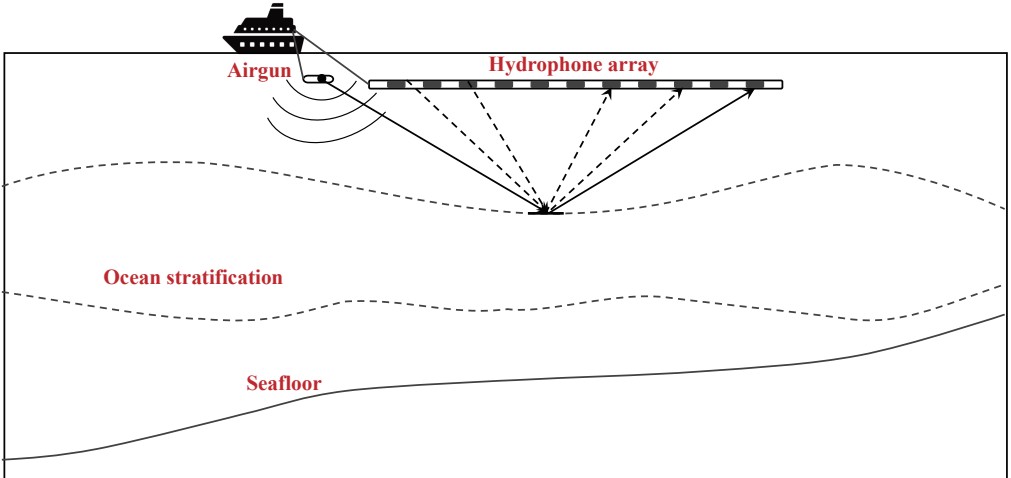

**Figure 7.** Demonstration of seismic data collection.

The ocean can be considered to be made up of uniform thin layers, the acoustic impedance of the water layer at depth of $z$ is defined as [69]:

$$I(z) = \rho(z)\, v(z) \tag{6}$$

here $\rho$ is the density and $v$ is the sound velocity. As is known, the $\rho$ and $v$ can be represented with the temperature, salinity and pressure by the equation of state. The reflectivity of $j$th water layer and the one immediately above is:

$$R_j = \frac{I_{j+1} - I_j}{I_{j+1} + I_j} \approx \frac{\Delta I}{2I} \tag{7}$$

The approximation in the above equation is reasonable as long as the thickness of each water layer is small enough. With the reflectivity, the seismic seismogram can be computed by the convolution of reflectivity with the source wavelet:

$$S = w * R = \frac{w * \Delta I}{2I} \tag{8}$$

here $w$ is the source wavelet. As can be figured out, with the observed seismogram to obtain the acoustic impedance to retrieve the salinity of the seawater is an inversion problem. Different inversion protocols have been developed to inverse the SO data to retrieve the temperature and salinity and will be described in the following sections.

### 4.2. Full Wave Inversion

The full wave inversion (FWI) is an iterative tool that improves the model by directly comparing synthetic seismograms with the observed seismograms by means of an objective function [64,70,71]. FWI is especially suited for ocean physical parameters retrieving with seismic data [72]. Since water can only support P-waves propagation, there is no converted shear waves present. Besides, the lateral sound speed variations are very small due to flat thermohaline boundaries in the ocean. Interbed multiples can be negligible due to small reflection coefficients. As shown in Figure 8, the general procedure of FWI includes data processing and a starting model, conducting the forward simulation, calculating and evaluating the residuals, and optimizing the simulated model to obtain the optimal model. With the inverted sound velocity, the temperature and salinity can be retrieved by the equation of state and empirical T-S relationship. The performance of different FWI approaches are compared in Table 5.

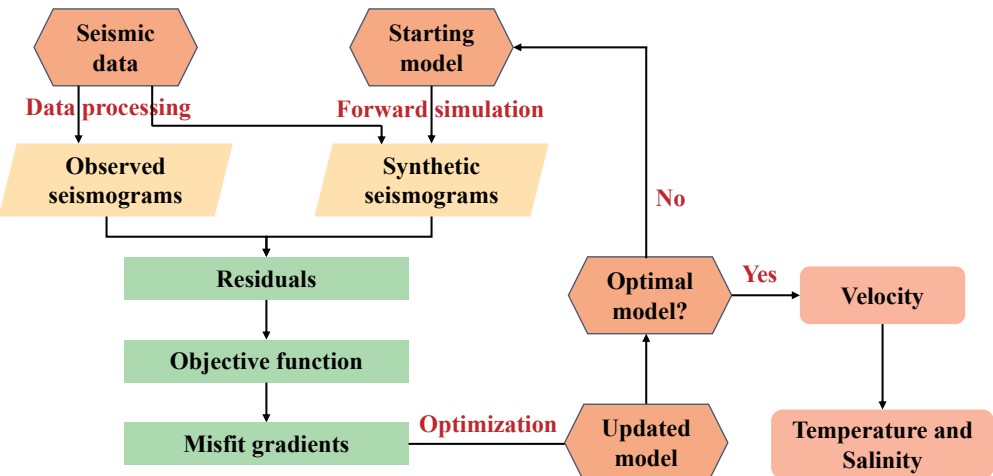

**Figure 8.** The workflow of FWI, reprinted with permission from Refs. [64,73]. Copyright (2022), with permission from Wiley and Nature.

Wood et al. firstly applied FWI to water layer and successfully resolved the temperature profile down to the water depth of 750 m [72]. Comparing the retrieved temperature data with the true temperature data collected by the XBT, the inversion accuracy was within one degree Celsius.

Kormann et al. applied the one dimensional FWI to retrieve the temperature and salinity of the water column [64]. The starting model was generated with the sound speed profile corresponding to the CTD profiles acquired in the field. The forward simulation was conducted with a time-domain finite difference scheme. Adopting the $L_2$ norm as the objection function, the optimization of misfit gradients was achieved with the adjoint method [74]. Because the contribution of the sound speed to the acoustic impedance of the water layer, i.e., the water reflectivity, was approximately 99%, only the sound speed was converted [69]. To determine the temperature and salinity from the inverted sound speed profile, the following two expressions were chosen: the sound speed formula that express $c$ as a function of $T$, $S$ and $z$ [75,76], and a neural network (NN) T-S relationship. The root mean square of the inverted physical properties was 0.1 °C for the temperature and 0.06 for the salinity.

The empirical T-S relationship may induce inversion errors since it is used for all the seismic section, taking no consideration of different mesoscale structures. A direct temperature and salinity acoustic full wave inversion approach was developed without the need of an empirical T-S relationship [77]. The inversion protocol was similar with that proposed by Kormann et al. [64], except that the gradients, or the sensitivity kernels of *T* and *S* were calculated via the adjoint method. Commonly, the kernels were derived for the parameters in the acoustic equations, it was *K* in the Ref. [64]. The root mean square (RMS) of the resolved salinity profile from this direct inversion method was 0.01, which was one-sixth of that obtained following the *K* strategy [64].

The FWI method has been tested by 1-D synthetic and observed seismic data, the feasibility of 2-D adjoint-state FWI of salinity and other water layer properties using prestack MCS data were also demonstrated [78].Since one potential difficulty of the 2-D FWI is the poor signal-to-noise ratio (SNR) of the individual traces, a specific data processing procedure was designed to reduce noise without modifying the recorded waveform. The starting model was built by linearly interpolating the 1-D velocity profiles given by the individual XBTs. The sound velocity was retrieved by the dedicated 2-D FWI based on the adjoint method and multiscale inversion strategy. *T* and *S* can be acquired by *v* with the following two equations. The first equation was the thermodynamic equation of seawater to represent *v* as a function of *T*, *S* and pressure *P* [20]. Another was the P-dependent T-S relation referenced from regional data of the National Oceanic and Atmospheric Administration (NOAA) database [79]. Without the simultaneously CTD measurements at the same location with the seismic acquisition, the inversion accuracy was estimated comparing with the CTD measures done by the German R/V Poseidon on average 2–3 h after the seismic acquisition which gave a standard deviation of 0.08.

**Table 5.** The comparison of different FWI approaches.

| Ref. | Starting Model | Misfit Gradient | Optimization | TS Inversion | Accuracy |
|------|----------------|-----------------|--------------|--------------|----------|
| Kormann et al. [64] | two real CTD profiles | Adjoint method | nonlinear Conjugated Gradient method | Sound speed formula; neutral network T-S relationship. | *T* (0.1 °C), *S* (0.06) |
| Wood et al. [72] | Low-pass filtered versions of the XBT profile | Reflectivity method | Newton's Method | Equation of state (assume salinity is constant) | *T* (0.5 °C) |
| Bornstein et al. [77] | real CTD cast | Adjoint method | iterative nonlinear conjugate gradients | Direct inversion | *T* (0.03 °C), *S* (0.01) |
| Dagnino et al. [78] | XBTs interpolation | Adjoint method | quasi-Newton limited-memory Broyden-Fletcher-Goldfarb-Shanno algorithm | Thermodynamic equation of seawater; P-dependent T-S relation inferred from the NOAA database | *T* (0.18 °C), *S* (0.08) |
| Padhi et al. [80] | GA-inverted | non-linear least squares inversion / GA | UNESCO algorithm; CTD data to obtain the regional T-S relationship | *T* (0.19 °C) / *T* (0.17 °C) |

Previous inversion methods that obtain sound velocity to resolve *T* and *S* are local inversions and their accuracy is strongly affected by the initial model. The initial model is typically built with the data from XBTs deployed along the seismic line. However, XBTs are not always available or the data are too limited to obtain an initial velocity model. The global 2-D seismic waveform inversion method based on the generic algorithm (GA) was proposed [80] without the need of XBT data. The initial model was generated by apply GA at sparse locations over a 2-D seismic survey to get the sound speed profile to start the inversion. The sound speed inversion followed two strategies for comparison between the local and global inversion method. One was the local non-linear least squares (NLS) inversion protocol, while the other was the global inversion with parallel implementation of GA. The method combined the GA to create the initial model and NLS for sound speed inversion was called hybrid method. The hybrid and GA method presented competent inversion results, the RMS of *T* from the former was 0.19 °C and the latter was 0.17 °C over the entire range of the inversion. However, the computational burden of the GA method was much larger than the hybrid method and the inversion time of the GA method was 18–20 times of the hybrid method. Although the global inversion technique can retrieve temperature and salinity profiles without supporting oceanographic information, the computation cost is worthy of concern.

### 4.3. Deconvolution Inversion

Another approach that enable to extract *v* and/or *T* and *S* out from acoustic reflectivity was pioneered by Papenberg et al. [81]. As shown in Equation (8), the seismic seismograph can be computed by the convolution of reflectivity with the source wavelet. By seismograms deconvolution, the wavelet was removed to obtain the reflectivity of the water layer [65,81,82]. Then the physical parameters of the water layer, including temperature and salinity, can be recovered accordingly. As a result, we call this method deconvolution inversion.

The typical workflow of the deconvolution inversion technique is shown in Figure 9. The data preconditioning is necessary to generate a true-amplitude time-migrated seismic section. The combined seismograms are deconvoluted with an estimated source wavelet to recover the reflection coefficients. Two strategies were proposed to retrieve the *T* and *S* with the gained reflection coefficient. Papenberg et al. [81] firstly converted the reflection coefficients to sound speed with short wavelength due to the band limited nature of the collected seismic data. A smooth long wavelength sound speed model was built based on the XBT and CTD casts acquired during the survey. Then assume the density was constant and started the iteration process by a constant salinity to initiate the sound speed. Until the error was less than the noise, the iteration ends and *T* and *S* were obtained. The inversion method was applied to the multi-channel seismic data acquired in the Gulf of Cadiz and retrieved the *T* and *S* profiles to depth of 1700 m with accuracy of 0.1 °C and 0.1 psu, respectively. Another strategy refers to invert the reflective coefficient to the acoustic impedance [65]. Different from Papenberg et al., the density was not considered as a constant, it was derived from inverted *T* and *S* instead. Moreover, the neural networks (NN) were used to retrieve salinity out of temperature. Biescas et al. compared the two strategies for temperature and salinity recovery from reflectivity and found that the former one performed better in the shallow (0–500 m) and deep (1500–2000 m) area, while the latter yielded much better results in between (500–1500 m). The accuracy of the inversion in Biescas et al. was 0.09 for *S*, slightly better than the former strategy.

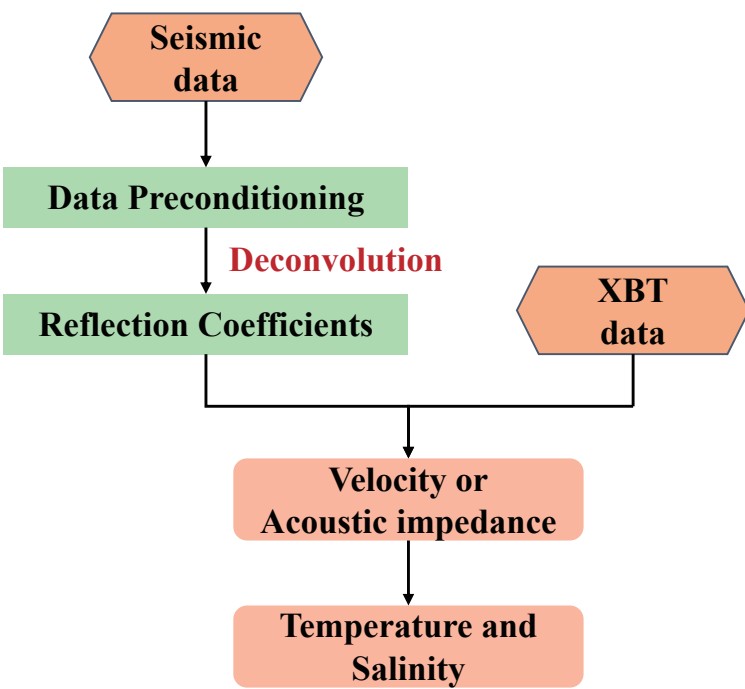

**Figure 9.** Typical workflow of the deconvolution inversion method.

*4.4. Markov Chain Monte Carlo inversion*

In previous seismic inversion methods, the inversion accuracy was estimated from the resolved data with the hydrographic observations. However, the offset of the observation time and position between the hydrographic data and the MCS data is inevitable. This will bring uncertainties that is difficult to quantify. To quantify this uncertainty, a Bayesian Markov Chain Monte Carlo (MCMC) approach was applied to invert the T-S structure from the seismic data [83,84]. Three steps were required to estimate *T* and *S* with quantified uncertainties: true-amplitude seismic data processing; salinity estimation from XBT data; and modeling using the MCMC approach. As can be found, this inversion strategy is similar with the deconvolution method [65,81], apart from the addition of MCMC approach. The new approach was then corroborated with the synthetic data computed from a CTD cast and the deconvolution method [65] to validate its feasibility. Based on the application of this approach to the acoustic reflectivity section, the salinity accuracy of 0.006 psu was better than the deconvolution method. For the first time, this method had enabled the detailed portray of the internal solitary wave (ISW).

Xiao et al. made further progress of the MCMC inversion method to improve the prior model by iterative updating [84]. The prior model was created based on the XBT data collected every 2.3 km and the salinity data estimated using the neural network approach [85] from the temperature data. At each inversion step, the prior model was iteratively updated to incorporate both the hydrographic data and previous inversion results. After application of this method to observed seismic data in the Gulf of Cadiz, the inversion uncertainties of the salinity averaged across the seismic section was 0.055. It was found that the main contributor to the final uncertainty was the low frequency starting model derived from the XBT interpolation. As a result, an accurate starting model is crucial for reducing the final inversion uncertainty.

*4.5. Geostatistical Seismic Inversion*

Statistical approaches are promising to solve seismic inverse problems in oil and gas reservoir characterization [86]. An iterative geostatistical inverse methodology was first applied to seismic oceanography data to retrieve reliable high-resolution temperature and salinity models [87]. The inversion procedure of this approach is shown in Figure 10. The model parameter space, i.e., the temperature and salinity, is perturbed using stochastic

sequential simulation and co-simulation. The reflective coefficients can be computed with *T* and *S* models according to the international thermodynamic equation of water [88]. With the extracted wavelet from the observed seismic data, the synthetic seismograms can be calculated. Then the mismatch between synthetic seismograms and the observed seismograms was calculated and the temperature and salinity traces with lowest misfit were selected as secondary variables in the co-simulation of temperature and salinity models. The iteration process stopped until the global coefficient between the real and synthetic seismic data was above a certain threshold $C_{th}$.

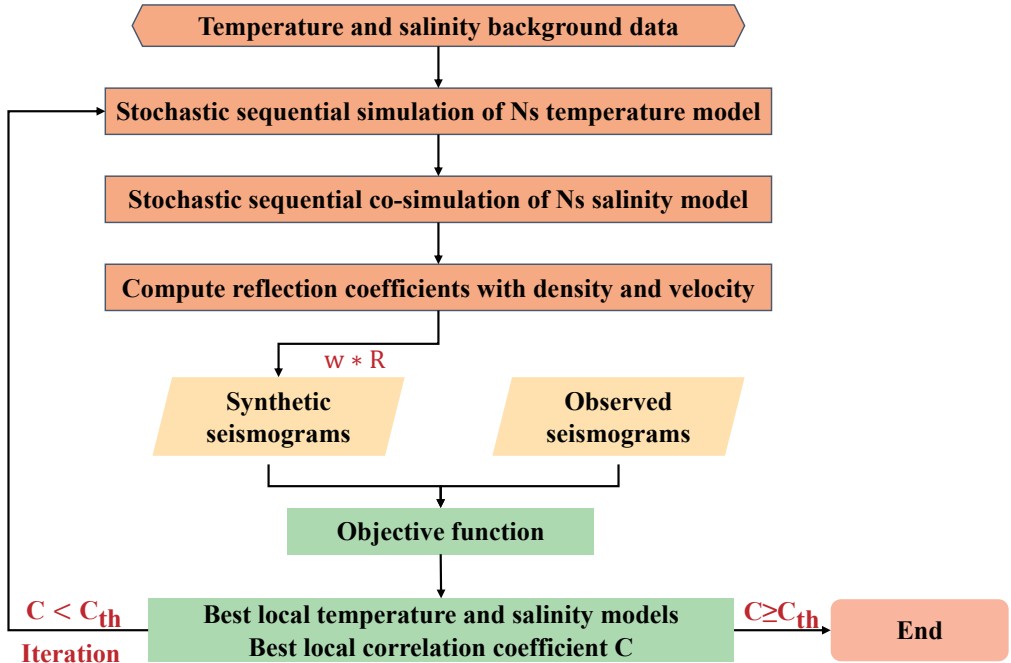

**Figure 10.** Typical workflow of the geostatistical seismic inversion method, *C* represents the global coefficient between the real and synthetic seismic data and $C_{th}$ is the threshold of the global coefficient. Adapted with permission from Refs. [87,89]. Copyright (2022), with permission from Springer and Frontiers.

In the preliminary application of iterative geostatistical inversion method in the seismic oceanography data, the temperature and salinity background data were directly measured simultaneously and collocated with the seismic data. However, simultaneous acquisition of CTD and XBT data collocated with the seismic data is challenging in practice. Azevedo et al. [89] proposed an alternative approach without the need of collocated and simultaneous temperature and salinity data. Instead, they adopted the data from global numerical simulations of ocean dynamics provided by the Copernicus Marine Environment Monitoring Service [90]. The inverted temperature and salinity models had higher resolution than the observed seismic data and enable to show the fine structure of the ocean. This is one of key advantages of using the geostatistical inversion method. Another benefit refers to its ability to assess the uncertainty associated with the model predictions. According to the comparison of the inverted two-dimensional sections of temperature and salinity with the vertical one-dimensional profiles acquired by the ARGO floats, the value of the inverted parameters deviated from the direct measurements while their main trends are reproduced. This deviation can be improved by additional constrains from other oceanographic variables.

## 5. Conclusions

The development of ocean salinity measurements is systematically reviewed in this paper and the brief comparison of different salinity measurement approaches can be

found in Table 6. The electronic sensors exhibit the highest measurement resolution of marine salinity and are the mostly used instruments in the oceanographic field. Future direction in this area is towards producing small and low power conductivity sensors that are satisfying the high demands on measurement uncertainty needed for oceanographic research. Another kind of direct observation technique to obtain the marine salinity is the optical methods based on refractive index measurement. This is a thriving area, while most of the sensors are still at the laboratory study stage. Further efforts should be made to improve the performance and robustness of these sensors, and finally achieve industrialization. The seismic observation methods is a newly developed field and a new branch, namely the seismic oceanography, is established accordingly. Great efforts should be paid to expand its application areas to make deeper understanding of the vast ocean.

**Table 6.** The comparison of different salinity measurement approaches.

| Approach | Measurement Object | Measurement Mode | Lateral Resolution | Vertical Resolution | Salinity Resolution | Technical Maturity |
|---|---|---|---|---|---|---|
| Electrical | electrical conductivity | direct | several km | ~1 m | $10^{-4}$–$10^{-3}$ | high |
| Optical | refractive index | direct | several km | ~1 m | 0.001–1 | low |
| Seismic observation | acoustic waves | indirect | ~10 m | ~10 m | 0.01–0.1 | modest |

**Author Contributions:** Conceptualization, L.G. and X.H.; investigation, L.G. and X.H.; data collection, L.G. and X.H.; writing—original draft preparation, L.G. and X.H.; writing—review and editing, M.Z. and H.L.; supervision, M.Z. and H.L.; funding acquisition, X.H., M.Z. and H.L. All authors have read and agreed to the published version of the manuscript.

**Funding:** This work was funded by the National Natural Science Foundation of China (NSFC) with grant No. 62105007; the financial support from China Geological Survey with grant No. DD20221703; and the Science and Technology Program of Guangzhou with grant No. 202103040003.

**Institutional Review Board Statement:** Not applicable.

**Informed Consent Statement:** Not applicable.

**Data Availability Statement:** Not applicable.

**Conflicts of Interest:** The authors declare no conflict of interest.

## Abbreviations

The following abbreviations are used in this manuscript:

| | |
|---|---|
| CTD | conductivity temperature depth |
| MEMS | Micro-Electro-Mechanical System |
| PCB | printed circuit board |
| PSD | position-sensitivity detector |
| TCF | two-core fiber |
| SMF | single-mode fiber |
| MMI | multi-mode interference |
| NCF | no-core fiber |
| ECF | exposed-core fiber |
| PCF | photonic crystal fiber |
| SI | Sagnac interferometer |
| PC | photonic crystal |
| FBG | fiber Bragg grating |

| | |
|---|---|
| PI | polyimide |
| FP | Fabry-Perot |
| PSU | Practical Salinity Unit |
| XBT | expendable bathythermographs |
| MCS | multichannel seismic |
| SO | seismic oceanography |
| FWI | full wave inversion |
| NN | neutral network |
| GA | generic algorithm |
| NLS | non-linear least squares |
| MCMC | Markov Chain Monte Carlo |
| ISW | internal solitary wave |

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
