# Peer review of "Advances in the Technologies for Marine Salinity Measurement"

_jmse, doi:10.3390/jmse10122024_

Round 1

Reviewer 1 Report

General remarks

In this review three categories of technologies for the determination of salinity of sea water are presented. 

The standard method for decades, the electrical conductivity, the optical method of determining refractive index of seawater as methods of point measurements and the method of seismic observation and inversion, akin to the acoustic tomography, yielding a bigger picture of the horizontal distribution of T and S.

In the section on electrical conductivity measurement there is no mention of the two different ways, conductance or inductance measurement.

In the section on optical methods there is a separation into three groups of refractive index measurement named as beam deviation, light wave mode coupling and surface coating. A total of 20 different articles on optical measurements are listed, how these were selected over others remains in the dark. 

While for the conductivity sensors numbers for range, resolution and accuracy are given in terms of conductivity there are no numbers for accuracy (or repeatability) for any of the optical methods. Sometimes reference is made to % of salinity, where it is not clear if that is meant in the sense of ‰ (i.e. 10 ‰ equal to 1 %) or in % of saturation concentration. Even the units for the measurement range are not always clear, they are given in ‰ or % or mol/L - why they are not given in terms of refractive index defeats me.

It would be much more useful for the reader if this review would concentrate on fewer refractive index sensing technologies but with a weight on what methods could be applied to in situ sensing, what are the most promising candidate technologies. The figures of merit should be given in a way that allows comparison to the sensors based on conductivity measurement. I do see the problem that the determined quantities (conductivity or refractive index) relate to salinity only indirectly and not without the influence of other parameters (like temperature and pressure).

Shortening the section on optical sensors would then give room to introduce the basic principles of conductivity measurement on one hand and the section on the use of seismic measurements to determine the T/S fields on the other hand. The latter method may not fit exactly with the other two as it is not a point measurement, nonetheless I see some value in raising awareness of this method in the oceanographic community. Within the scope of this manuscript I feel, however, that this section could be shortened and more emphasis laid on the sections on in situ sensors.

More detailed comments and remarks

Introduction line 17-18

The definition given here is not the one of the cited publication of Lewis but rather is a citation from within that publication of the definition given by Forsch, Knudsen and Sorensen in 1902 (C. Forsch, M. Knudsen and S.P.Sorensen, "Reports on the determination of the constants for compilation of hydrographic tables", Kgl. Dan. Vidensk, Selsk, Skifter, 6 Raekke Naturvidensk. Mat. Copenhagen, vol. 12(1), pp. 1-151, 1902.)

The publication of Edward Lewis (reference [2] of this manuscript) intended to explain the then new definition according to the practical salinity scale of 1978 (PSS-78) "The basis for this new scale is an equation relating the ratio of the electrical conductivity of the seawater sample to that of a standard potassium chloride solution (KCl) at 15°C atmospheric pressure".

This definition of salinity has since been replaced (or at least complemented) by that of the absolute salinity in the TEOS-10.

(as e.g. published in "The international thermodynamic equation of seawater - 2010: Calculation and use of thermodynamic properties", Intergovernmental Oceanographic Commission, Manuals and Guides 56) https://www.teos-10.org/pubs/TEOS-10_Manual.pdf

line 20-21

A unit psu does not exist! The practical salinity is also not given in ppt or ‰. The TEOS-10 manual (see last comment) states: "Note that Practical Salinity is a unit-less quantity. Though sometimes convenient, it is technically incorrect to quote Practical Salinity in “psu”; rather it should be quoted as a certain Practical Salinity “on the Practical Salinity Scale PSS-­78”." The practical salinity, as based on a conductivity ratio, has no unit. Thus the mentions of "psu" in this manuscript should be avoided. 

The absolute salinity, as defined in the TEOS-10, however, does have the unit of g/kg. It would be worthwhile for the authors to read the section on the practical salinity and absolute salinity in the TEOS-10 manual and change the parts in this manuscript accordingly. It probably also would be instructive in an overview paper like this to make sure that the readers understand that psu is not a unit and practical salinity is without unit.

line 27

I wouldn't really describe the 70's conductivity sensors as mature. There was still a long way ahead to arrive at todays conductivity sensors that fulfil the high oceanographic standards. 

line 29-31

Bulky, expensive and high power are relative. While there are applications where bulk and power consumption of 100 mW may pose problems in most research applications they are not the most pressing problems. The difficulty is to produce small and low power conductivity sensors that are satisfying the high demands on measurement uncertainty needed for oceanographic research.

line 31-33

I don't see how the reference [9] relates to seawater conductivity measurements. Also how do particles in the seawater "interact with the ionic compositions"? I do concur, though, that the conductivity measurement has its shortcomings, especially when the salinity measurement is only an intermediary for the determination of density.

line 36-38

Both sensor types, electrical conductivity sensors as well as refractive index sensors exhibit some measurement uncertainties. Both types of sensors are not measuring the salinity (in the sense of the definition of Forsch, Knudsen and Sorensen) but either the electrical conductivity (or a ratio of conductivities) or the refractive index (that is more directly tied to the density of the water than to the salinity). 

(Even before the introduction of the PSS-78 refractometers were used to determine density and/or salinity of seawater samples.)

line 38-40

Refractive index sensors don't "consider" anything. They are instruments to determine the bulk refractive index of the measurement medium (i.e. seawater in our case). The bulk refractive index is including all substances dissolved in the measurement medium, so all of those are contained in a salinity measurement based on the measurement of refractive index.

I'm not sure that a new definition of salinity is formed (if anything, the definition is more closely related to the old one of Forsch, Knudsen and Sorensen). It is perhaps unfortunate to call either measurement a salinity measurement. Neither electrical conductivity nor the refractive index is the same as salinity. A sugar solution would result in a salinity value when measured via refractive index, but not if measured via conductivity. But I guess that is more a philosophical problem and shouldn't be considered at length in this manuscript.

line 45-50

Hydrography not only endeavours to describe the physical features, it is more geared towards an understanding of the circulation on the various temporal and spatial scales. It is by no means restricted to take vertical profiles from ships, these days there is a variety of platforms like AUVs, gliders, ARGO floats, autonomous surface vehicles,... that operate on various spatial and temporal scales. Surface salinity measurements from space complement the picture. Yes, the spatial and temporal resolution is very limited but it's a lot more versatile than taking profiles from ships.

In general I'm not sure if the inclusion of the seismic oceanography part does make this manuscript stronger or if it makes it too broad. I like the inclusion as it is a method not really in the mind of many oceanographers, but on the other hand it is so completely different from the point measurements of conductivity or refractive index sensors.

2. Salinity measurement with electronic sensors

(I would rather rename this section "salinity determination using electrical conductivity sensors") as refractive index sensors could also be classed as electronic sensors as they employ a lot of electronics as well.

line 69-70

As mentioned above the practical salinity does not have a unit, I would write "where S is the practical salinity". The decibar is written as dbar, not dBar (and nowadays pressure should be presented in SI units - I do see that this wouldn't be very practical here, so the usage of dbar is tolerable).

line 73

It is worth noting that the stated conductivity of 42.914 S/m is not part of the definition of the practical salinity scale PSS-78. It is stated there for informational purposes only. 

In general it would be preferable to state the conductivities in S/m (as this is the proper SI unit) rather than in mS/cm. 

line 79-80

Then we use eq. (1) to (4) to show the relation of salinity and conductivity, temperature and pressure.

figure 1

practical salinity without unit, dbar instead of dBar and preferably conductivity in S/m.

Generally the axes should be labeled by variable / unit (e.g. T / °C ) as suggested by the SI brochure. (This way you can read the graph as  variable divided by unit equals #, e.g. T / °C = 20 or, if you rewrite that equation, T = 20 °C) (same holds true for table 1 as well)

For the caption: It's not a simulation.

table 1

What is meant here by "sensitivity between salinity and conductivity"? Is that the ratio of change of salinity to change of conductivity at the temperature-pressure points?

If so, it would be good to add a Delta S/Delta C or dS/dC to that column.

table 2

I think there is a 0 too many in the resolution for the NKE conductivity sensor.

line 102

The CTD does not measure depth, it measures pressure (from which then the depth is inferred). It's similar to the conductivity measurement that is used to infer salinity.

line 105

Write "CTD instruments have been developed to maturity, ...". It may be worth noting the two different concepts for conductivity measurement, the multiple-electrode cell and the inductive cell (I think RBR is using an inductive sensor, Valeport as well, not sure about Aanderaa)

line 106-109

What is your reference for the claim of "bulky, expensive and high power consumption"? Without giving numbers this can mean different things for different people. Compared to a wave radar the power consumption is low, as is the bulk of a CTD. What is driving the miniaturization? What applications are suffering from the bulk of CTDs?

There are also miniature CTDs around, e.g. a system build by Valeport for the Sea Mammal Research Unit in St. Andrews (see e.g. Biuw et al. 2007 doi:10.1073/pnas.0701121104 ). They are using a small, but conventional conductivity cell successfully in polar regions, operated of a D-cell battery for some months (so hardly high power consumption).

line 110-126

Building a compact sensor is nice, but where is a salinity measurement uncertainty of 0.5 sufficient?

How does the power consumption compare to the commercial CTDs? RBRs legato has a power consumption of 46 mW when sampling at 2 Hz, an energy demand of 22.8 mJ per sample.

(And, of course, delete the ‰ and use dbar)

The sensor of Wu et al. looks more promising but wasn't tested under pressure as far as I can see. There are also other developments (e.g. Huang et al. 2011, doi: 10.1109/JSEN.2011.2149516 ) that have been tested in the sea. What made you choose the ones presented here?

Figure 2

Have you obtained permission for reproduction of that figure? This comment also relates to other figures in this manuscript reproduced from other articles.

line 127-149

The electromagnetic interference of an underwater sensor with a metal pressure housing around the electronics is rather limited. So where does that pose a real world problem?

The external field problem described in Ashokan et al. does exist for the inductive sensors only - many of the commercial sensors are based on 4 (or more) electrode cells, having almost no external field. This is nothing special. 

What does it mean when in line 137/138 it is stated that the cell had an estimated (!) accuracy of +/-1.47 %? Is that 1.47 % of full scale? 

In line 147 there is an s missing in sensor.

It's not clear to the reader what is meant by "provide more snapshots than any other expendable CTD instruments". 

line 150-155

I don't see that generally "flexible materials have several advantages over rigid materials" nor do I see that flexible materials have "better mechanical and thermal properties". 

The last sentence (line 154/155) "While ..." has no end and thus no meaning.

3. Salinity measurement with optical sensors

(I would rather rename this section "salinity determination using optical sensors" (see comment on section 2))

line 158/159 "The optical instruments obtain the salinity value by measuring the refraction index of the seawater" It's refractive index, not refraction index. And you chose those optical sensors that measure the refractive index - other methods are possible as well (albeit with limited resolution in salinity).

line 159-162

I suggest to put the chain of arguments differently: 

1. the instruments measure refractive index

2. refractive index and density are closely related by Lorentz-Lorenz

3. density (and thus refractive index) changes with changes in S,T,p and lambda.

line 162-166

The reference is Millard and Seaver, not Milliar and Seaver (correct naming in the references). It is the accuracy of that algorithm varying it's accuracy with pressure, not the refractive index itself. Again, practical salinity has no unit, pressure has the unit dbar (of course you could convert that to the SI unit of Pa as well).

line 174

We use eq. (5) to show the changes of refractive index with changing salinity, temperature and pressure. (It's not depth!)

figure 3

Basically same comments as for figure 1

table 3

Again, what is the sensitivity? Is it Delta S/Delta n (or dS/dn)? 

line 187-188

The method of measurement of the refractive index doesn't matter (though I would think that optical means are preferable). To obtain the salinity you also need to know temperature and wavelength (and pressure), though.

line 188-189

While it is true that there are many optical methods still in development (and have been since the 1980s) there is also a product in the market from NKE https://nke-instrumentation.com/produit/noss/ This should at least be mentioned in the manuscript.

line 191

Surface coating in itself is not a measurement method. What do you mean by the term "structure configuration"?

line 201

There are earlier examples like the ones referenced in Millard and Seaver (your ref [39])

by Mahrt and Kroebel in 1984 (a bench salinometer based on an interferometer), Seaver in 1986 or Mahrt and Waldmann in 1988 (with field data), the latter two being refractometers.

That leads to the question how you selected the examples to be included in this manuscript.

line 222-225

Yes, that's the nature of a PSD to give the position independent of intensity. It's all well to know the resolution of salinity measurement, however, it would also be good to get a figure for accuracy (or at least repeatability) for the salinity determination. (And, of course, salinity should be given without units here again.)

line 241-243

Isn't it rather that a part of the evanescent field is coupled as a plasmon in the metallic layer and thus the transmitted light is reduced?  

line 252

No, if the refractive index difference between core and cladding decreases then the light transmitted should be getting less. If not then a fibre with core and cladding of the same refractive index should guide light best.

line 2533-254

What is meant by a sensitivity of mV/%? Why is the linearity so important? And what salinity is a solution concentration of 12 % ? Is that supposed to be a salinity of 120? Or is it 12 % of saturation concentration?

line 254-261

The same questions as before, what are the % denoting? Was there any investigation on how mechanical disturbance or pressure would act on this sensor? What accuracy or repeatability could be reached in the real world with such a sensor? 

figure 6

The first question here again is, if you obtained permission to reproduce the figures from other articles here. Figure 6 (a) is very basic and doesn't explain a lot unless the black oval shape is supposed to show what can be seen in panels (b) and (d). 

line 262-264

When the intensity is depending on the refractive index difference between core of the fibre and the seawater as the cladding (or on the evanescent field partly escaping the fibre like in the case of the multi-mode plastic fibre), then the absorption due to turbidity is not affecting the guided light that remains in the fibre. So it is worthwhile looking into what else can change the intensity of the light transmitted through the fibre, as I agree that it most probably is not a good way of measuring salinity. But turbidity is not what interferes with the measurement.

line 265-275

I think I haven't understood how this is supposed to work. The image in fig. 6 (a) (the square inlay) suggests that there is cladding around both cores, around 40 µm to 50 µm minimum around the off-centre core. How is that supposed to sense the medium outside the fibre? The evanescent field reaching out of that fibre should be pretty weak already. Or is it the case that the core is very close to the outside medium as suggested by the lower part of figure 6 (a)? If it was the latter then I could understand the basic mechanism of the influence of the surrounding medium. What I do not understand, however, is how the single mode fibre does transmit a broad spectrum of wavelengths as suggested by fig. 6 (g), showing high transmission at around 1525 nm as well as at 1612 nm. It should also be mentioned, that this set-up needs a spectrometer (or spectrum analyzer) at the receiving end to determine the position of the resulting dip. 

The given figures should be converted to salinity to be comparable with the other sensors. 1 mol/l NaCl would be 58.44 g/l, so roughly a salinity of 58.44, making the sensitivity about 0.24 nm (per salinity unit). To get to CTD standards of about 0.002 in salinity it needs to resolve around 5 pm. The range cited of 0 to 5 mol/L NaCl is a range of about 0 to 292 in salinity.

line 277-279

That sentence is missing a part. 

line 281-286

What is "the effective refractive index of the fundamental mode of the NCF"? 

line 286-289

Again, what is the percentage that is related to here? Can you please give these specifications in terms of salinity (even if you have to make the assumption that g/l NaCl equals salinity the error would be not too big and the comparison of the different sensors would be made much easier).

line 290-299

If I understand correctly this sensor measures the refractive index of the surrounding medium. It is termed all fibre CTD but it isn't explained how the influences of the salinity, temperature and pressure can be distinguished, as at the end of the day it is an all fibre refractive index sensor and not a CTD. 

line 309-315

Again the question is, what is meant by 100 % salinity. It certainly is not - as suggested in your table 4 - a salinity of 1000, as that would by far exceed the solubility of salt in water.

Please convert the units of all these sensors to practical salinity (or absolute salinity). For this paper it should be possible using the figures of merit in terms of refractive index and the algorithm from Millard and Seaver or similar.

317-332

Are you really suggesting that any of the referenced sensors are candidates for an in situ salinity sensor? 

Section 3.4

The same that has been said already applies here as well. Please make the numbers transparent and comparable. Sensitivity doesn't say much in the presence of noise. To compare sensors the accuracy or at least the repeatability are needed.

The swelling of a coating in water as a sensing mechanism sounds very problematic to me, as here, once again, the refractive index of the coating material is what is measured. Cross sensitivities (e.g. from hydrophobic substances in the sea water accumulating in the coating material) could pose a serious problem. 

What is sorely missing from this list of different refractive index sensors are straight forward refractometers, like the one available from NKE commercially or the one from Mahrt and Waldmann 1988 (see above) or Waldmann and Thiele 1996 ( doi:10.1109/OCEANS.1996.569061 ). I would think that such sensors are much better candidates for optical in situ salinity (or rather density) sensors. 

Table 4

As mentioned before this table is only useful to the reader if and when it contains some numbers on accuracy or at least repeatability that can be achieved with these sensors. An idea of the measurement frequency would be quite useful as well, it would show that the sensors based on the swelling due to water content of a coating are disqualified for profiling measurements and would be of interest for e.g. moored applications only.

Section 4 - Salinity measurement with seismic observation

The section is generally well written but perhaps a little too detailed for the remit of this article. It could perhaps be shortened to present this method as a supplement method to the more precise point measurements, revealing the horizontal spatial structure and thus helping the interpretation of the vertical measurements with in situ sensors.

Reviewer 2 Report

Paper is written in a very good way. Results are shown clearly and the images describe completely the scientific background and the goals that you want to obtain. But, according to wavelet, I have not found references to Torrence and Compo Wavelet program. This is a milestone in scientific use of Wavelet. I suggest these two references:
1) G. Pucciarelli, Wavelet Analysis in Volcanology: the case of Phlegrean Fields, Journal of Science Environmental and Engineering A, Vol.A, No.6, pp. 300-307, 2017
2) C. Torrence, G.P Compo, A Practical Guide to Wavelet Analysis, Bulletin of the American Meteorological Society, Vol. 79, No.1, pp.61-78, 1998

Reviewer 3 Report

The paper is very interesting and  consistent with the aims of journal. There are some minor commets:

1. there is no the reference in line 163

2. Fig. 10 - what does mean the second dashed line? 

My recomendation is accept

Round 2

Reviewer 1 Report

I'm under the impression that my recommendation of major revision hasn't really registered and only the detailed comments and/or suggestions have been reacted on (i.e. like a minor revision). No reconsideration of the choices of what developments are presented has been conducted.

I think the authors should think about their target audience for this paper first. Then they should carefully consider if the examples chosen are the most relevant for that target audience.

I have made a few specific comments again but I would hope that the authors not only answer these by changes to the text (if and when they agree - disagreement always is an option), but take the whole paper structure and the chosen examples into consideration. If they still think this is the right choice for the target audience then so be it. I would be interested to learn what target audience this would be. It may well be that it is too far removed from my interests to see it.

I still think the concept of this paper works well when the content is focussed on those things that would benefit the considered target audience most. 

Some more detailed comments:

line 3

suggest to write "...salinity measurements in hydrographic observations is to use a standard conductivity-temperature-depth probe where the salinity determination is based on a measurement of electrical conductivity."

line 4/5

I disagree with that statement. The techniques in the papers presented may ultimately lead to a miniaturization and/or integration into CTD instruments, most are concerned with the presentation of the respective technology and are quite far away from incorporation into sea going instrumentation (especially that concerned with optical methods). I think that a more precise statement would be "This article describes some developments of recent years that could lead to a new generation of instruments for the determination of salinity in seawater." Yes, there are some references that were concerned with miniaturization (e.g. references [9],[35],[36]), mainly considering electrode systems, but most papers are presenting different technologies that may at some point in the future lead to in situ instruments, some of them also showing the potential for miniature salinity sensors. I cannot make out that the common aim of the presented papers are "research efforts in miniaturization and integration of CTD sensors".

That said I do think that the presented developments are interesting and that there is some value in having an article showing a cross section of these developments. 

line 5/6

I guess you are referring to the conductivity sensor (both conductivity as well as optical sensor are incorporating electronics). Gas bubbles by definition are not dissolved. And dissolved gasses also affect the refractive index as they also affect bulk density of the water. The interest in refractive index sensors does originate more in the problems of traceability for the salinity determination using electrical conductivity sensors than in the susceptibility to "suspended ion particles" (I guess what is meant here is the net charge of suspended particles) or dissolved gasses.

line 10/11

Suggest to write "Complementary to the direct measurement salinity point sensors..." The sensors are measuring only at one point and thus don't have any lateral resolution. What determines the lateral resolution is on what platform they are deployed and what movement in space and time this platform describes. 

line 20/21

Carefully consider what definition of salinity you are referring to. The definition you are giving in line 17-20 is not entirely the same as Absolute Salinity. The relation of both those definitions to the practical salinity (as derived from a conductivity ratio) is another topic. You should get this sorted in your head (I'm not entirely sure that I got it sorted in my head, to be honest) and then think what the reader of this paper needs to know about these complications and what you can refer to other articles. 

As I see it the conductivity measurement does lead to the practical salinity while the determination of salinity by measuring the refractive index leads to something more closely related to the absolute salinity.

I don't know how to resolve this issue sufficiently well without going into the details. A way forward may be to name the difference between practical salinity and absolute salinity, mention that conductivity is more closely linked to the former and optical determination linked to the latter and refer the reader for details to the TEOS-10.

Just saying "The Absolute Salinity, as defined in the TEOS-10 [3], has the unit of g/kg." is not elucidating matters but may puzzle the unsuspecting reader. Why was the Absolute Salinity introduced? What are the differences between Absolute Salinity and practical salinity? Why do we care?

Maybe this paper is helpful in getting a clearer picture:

Millero et al."The composition of Standard Seawater and the definition of the Reference-Composition Salinty Scale", Deep-Sea research I 55 (2008), pp. 50-72, doi:10.1016/j.dsr.2007.10.001

line 27/28

Suggest to write "Sea water salinity was determined in the early 20th century using the chlorinity and the empirical Knudsen relation [10]."

As it is you jump through the timeline in the introduction. You start with the original definition, then jump to the latest definition of the TEOS-10, then back to PSS78 and in line 27 back to the Knudsen relation.

Perhaps you should consider to start the introduction with the way from the full chemical analysis via the chlorinity as suggested by Knudsen, the introduction of Copenhagen standard water, the electrical conductivity and PSS78 and finally the Absolute Salinity of the TEOS-10. 

So you could keep lines 17 to 19 and then rewrite after that. You could then mention that practical salinity has no unit and that Absolute Salinity has unit g/kg.

line 31

As in the first review I have to ask again: What is your metric for "bulky", for "expensive" and for "high-power consumption"? If you want to measure salinity at a certain rate, what is the energy needed to get one data point? What energy is needed for the same data point with some of the technologies you present in this paper? What is the estimate when that technology is matured and can be deployed in situ? How do these two measures compare?

(Looking at the example of the RBR concerto3 CTD being capable to run for 5 years at 1 sample per minute (2.6 million samples) on 8 AA batteries (about 36 Wh or 129.6 kJ) this results in an energy of 50 mJ per sample, which also includes the energy needed to store the data in memory as well as driving the real time clock. 50 mJ will light a LED with a forward voltage of 2 V at a current of 25 mA for 1 s, to get a comparison from the optoeletronics field.)

Same applies to the "bulk" argument, the RBR concerto3 is 440 mm long with a diameter of about 60 mm, so a volume of about 1.1 litres containing the three sensors, electronics and the AA batteries in a pressure housing for up to 6000 m. The conductivity sensor itself just takes up a tiny fraction of that volume. So be careful that you compare like for like.

It doesn't mean that there is no point in developing optical sensors and there certainly are applications where smaller size and lower power requirements would be very welcome (e.g. on gliders). 

line 33/34

Yes, of course the ionic composition of everything in the seawater will interact with the electric field, the ions in the water are what makes it conductive in the first place. It will also affect the bulk density and thus the refractive index.

line 36/37

How can dissolved gasses influence the salinity when it is defined, as you write in line 18/19 as "the total amount of _solid_ material in grams contained in one kilogram of seawater..."? The dissolved gasses are not solid material.

line 38-41

The refractive index sensor is not considering anything, it is not a thinking organism. I think it is clear what you mean but as it is written here it doesn't make sense. This was mentioned in the first review already.

line 41

What definition of salinity would that be? How is it different from the aforementioned definitions? If it really is a new definition the reader would need to know what it is.

line 44/45

I don't see how the fibre optic sensors are different from, say, refractometers in regard of pollution, corrosion resistance or electromagnetic interference. Also I wouldn't term fibre optic sensors as remote sensing, as they sense the medium in direct contact with the fibre. It would be in no way different from a conductivity cell on a cable.

line 49-52

This was also mentioned in the first review: The lateral resolution does depend on the platform you put the sensor on, not the sensor itself. The vertical profile is only one example. I guess what you are trying to convey here is that this type of sensor will always sample a single line in space and time whereas the determination using seismics gives a picture of a whole volume of water.

line 69-72

Yes, but that is no different for the determination of salinity from measuring the refractive index. If temperature, pressure and wavelength of the light are not known at the same time then you cannot determine the salinity from the measurement of refractive index.

line 137-143

Certainly the sensor presented by Aravamudhan et al. is interesting. But it seems a bit strange to point out a potential problem of inductive conductivity sensors as a comparison - you could have compared it to a standard conductivity cell that also has no problems with external fields. I fear that the additions you made here are triggered by my comment. I rather would revert these additions and just let the examples of Aravamudhan et al. and Ashokan et al. stand as examples of interesting developments without reference to weaknesses of one special type of standard conductivity sensor.

line 167-171

I keep up my comment from the last review:

"I suggest to put the chain of arguments differently:

1. the instruments measure refractive index

2. refractive index and density are closely related by Lorentz-Lorenz

3. density (and thus refractive index) changes with changes in S,T,p and lambda.

Response: Thanks for the reviewer’s suggestion, we have modified the description. (line 169-171)"

You didn't change anything on that sentence.

line 198

It's all very well adding that NKE sensor here. But considering that it is a method that has made it to a production instrument wouldn't it be appropriate to describe the principle of its operation here as one of the methods? Wouldn't it be the better example of a refractometer (beam deviation instrument)? And wouldn't it also be instructive to add it to table 4?

I don't understand why the choice of presented instruments/methods does not include the ones that are field proven.

To my earlier question regarding the choice of sensor articles you wrote: "As a review paper, we choose appropriate references to clearly describe the sensor system, measurement methods and effects as much as possible."

So how did you measure the appropriateness? How did you decide which development to include and which to leave out?

Table 4
On my remark that the table should give comparable figures of merit you answered "As a review paper, we can only list the parameters directly given in the references or calculated. We have converted the parameters list in the table to be comparable. (Tab. 4)"

But then I have to ask who the audience of this paper is supposed to be. What does that table tell the reader? Does one look at the sensitivity column and say the instrumental solution of reference 48 must be much better as that of 47 because the sensitivity in terms of voltage signal is much better? And the same for the wavelength specs, reference 59 must be much better as reference 60 as the sensitivity is 155.87 nm in the former and only 0.013 nm in the latter. 

How does the reader compare a sensitivity of 2.39 MHz to that of 4.2 mV or 20 nm?

If the technology used to determine salinity cannot be rated in terms of salinity then what does it tell us?

I assume that the sensitivity column is showing sensitivity per salinity unit, right? As these instruments measure refractive index mostly it could be also refractive index (as both are quantities with no unit). I suggest that you add this in the header "Sensitivity per salinity unit" or, if it isn't that, "Sensitivity per refractive index unit".

I do understand that this is not an easy undertaking but if you want to make this table useful for the reader you have to think how you arrive at some estimates for resolution and/or accuracy of these sensor concepts in terms of salinity. You may denote them as your estimates from studying the respective papers. I do think that this table is the centerpiece of your overview of different optical technologies for the determination of refractive index/salinity.

Round 3

Reviewer 1 Report

The manuscript has improved considerably from the first version and also from the second version. I have only a few minor suggestions as follows:

line 52
The density doesn't change with wavelength, it's only the refractive index that changes (and that change in refractive index will then result in the same density using the Lorentz-Lorenz relation ). I have looked at my earlier review and see that I may have introduced this error myself by a comment to the lines 167-171 in the second review that wasn't carefully worded and contains the same wrong statement. I'm sorry about that.  

line 58-59
Strictly that is not true for all developments. Mahrt & Waldmann 1996 ("Results of the dynamical tests of a special designed optical microstructure density probe based on the measurement of the refractive index") built an instrument with a resolution of 5E-7 and precision of 1E-6 in refractive index.
I would suggest to write "Up until now, the salinity measurement resolution of optical sensors is generally lower than that of conductivity sensors".

line 185
See comment regarding line 52.

line 252
Write "of reflection type" instead of "refection type".

Thanks for keeping up with my comments and suggestions that always were made in a constructive intent. I hope you share the feeling that the manuscript has improved considerably through the review process. 
